# Focused Ultrasound-Mediated Blood–Brain Barrier Opening Best Promotes Neuroimmunomodulation through Brain Macrophage Redistribution

**Alina R. Kline-Schoder [1], Rebecca L. Noel [1], Hemali Phatnani [2], Vilas Menon [3] and Elisa E. Konofagou [1,4,*]**

1.  Department of Biomedical Engineering, Columbia University, New York, NY 10027, USA
2.  New York Genome Center, New York, NY 10013, USA
3.  Department of Neurology, Columbia University, New York, NY 10027, USA
4.  Department of Radiology, Columbia University, New York, NY 10027, USA
*   Correspondence: ek2191@columbia.edu

**Abstract:** Neuroimmunomodulation is a promising form of drug-free treatment for neurological diseases ranging from Alzheimer's disease to depression. The evidence supporting the efficacy of focused ultrasound (FUS) neuroimmunomodulation is encouraging; however, the method has yet to be standardized, and its mechanism remains poorly understood. Methods of FUS neuroimmunomodulation can be categorized into three paradigms based on the parameters used. In the first paradigm, focused ultrasound blood–brain barrier opening (FUS-BBBO) combines FUS with microbubbles (MB) to transiently and safely induce BBB opening. In the second paradigm, focused ultrasound neuromodulation (FUS-N) harnesses the acoustic effects of FUS alone (without MB). In the third paradigm, focused ultrasound with microbubbles without BBBO (FUS + MB) combines MB with FUS below the BBBO pressure threshold—harnessing the mechanical effects of FUS without opening the barrier. Due to the recent evidence of brain macrophage modulation in response to FUS-BBBO, we provide the first direct comparison of brain macrophage modulation between all three paradigms both in the presence and absence of Alzheimer's disease (AD) pathology. Flow cytometry and single-cell sequencing are employed to identify FUS-BBBO as the FUS paradigm, which maximizes brain macrophage modulation, including an increase in the population of neuroprotective, disease-associated microglia and direct correlation between treatment cavitation dose and brain macrophage phagocytosis. Next, we combine spatial and single-cell transcriptomics with immunohistochemical validation to provide the first characterization of brain macrophage distribution in response to FUS-BBBO. Given their relevance within neurodegeneration and perturbation response, we emphasize the analysis of three brain macrophage populations—disease- and interferon-associated microglia and central-nervous-system-associated macrophages. We find and validate the redistribution of each population with an overall trend toward increased interaction with the brain–cerebrospinal fluid barrier (BCSFB) after FUS-BBBO, an effect that is found to be more pronounced in the presence of disease pathology. This study addresses the prior lack of FUS neuroimmunomodulation paradigm optimization and mechanism characterization, identifying that FUS-BBBO best modulates brain macrophage response via complex redistribution.

**Keywords:** microglia; central nervous system-associated macrophages; ultrasound; blood–brain barrier

## 1. Introduction

Focused ultrasound (FUS) has shown promise as a neuroimmunotherapeutic in a number of treatment paradigms. The most well-studied paradigm of FUS neuroimmunomodulation is focused ultrasound blood–brain barrier opening (FUS-BBBO), which combines FUS with intravenously injected microbubbles (MB) to transiently open the blood–brain barrier (BBB). FUS-BBBO was originally designed as a method of drug delivery. However, more

recently, it has been proven efficacious as a neuroimmunotherapeutic, ameliorating neurodegenerative disease pathology and cognitive deficits [1–9]. The second method of FUS neuroimmunotherapy is the combination of MB and FUS below the pressure threshold for BBBO (FUS + MB). FUS + MB has been shown to trigger preliminary neuroimmune system activation and has proven efficacious as a method of immunotherapy within the peripheral nervous system, particularly in the pancreas and the kidney [10–12]. The MB response to FUS in both FUS-BBBO and FUS + MB is best quantified by the stable cavitation dose (SCD). Reports have demonstrated that safe treatment occurs within a specific cavitation range, but within this range, higher SCD correlates with increased neuroinflammation [13–15]. Recently, FUS neuromodulation (FUS-N) has also been identified to reduce disease pathology and ameliorate cognitive deficits in models of neurodegenerative disease [16–18]. FUS-N utilizes FUS parameters that alter neuronal connectivity via a combination of mechanosensitive receptor interactions and transient hypothermia without the injection of MB [16,19].

Brain macrophages have been established as complex and heterogeneous cells with subpopulations varying with brain region, age, and disease [20–23]. They react in several ways to perturbation of the central nervous system by producing and emitting chemokines and cytokines, recruiting immune cells to the region, and removing toxic elements [21,24,25]. Two categories of brain macrophages exist—microglia, which are the brain's resident macrophages, and central-nervous-system-associated macrophages (CAM), which have a peripheral origin and primarily exist on and around the BBB and brain–cerebrospinal fluid barrier (BCSFB) [22,26,27]. disease=associated microglia (DAM) and interferon-associated microglia (IAM) are two subpopulations of microglia that are documented to respond to neurodegeneration and perturbation by surrounding the area of disturbance and phagocytosing debris and toxins (including neurodegeneration pathology). DAM were first identified in the context of Alzheimer's Disease (AD), where the conversion of homeostatic microglia into DAM was shown to be triggered by the reception of molecular patterns via receptors, including TREM2 and purinergic receptors [20,28]. Since their original discovery, DAM have been shown to be a neuroprotective population that contains and removes neuronal damage and pathology in both perturbation response and neurodegeneration. The role of IAM within neurodegeneration and perturbation is less clear—the population size has been shown consistently to increase in response to both neurodegeneration and perturbation, and recent evidence suggests that this population has a key role in neuronal pruning and debris removal via increased phagocytosis [29].

While single-cell transcriptomics allows for the identification of cell-type subpopulations, spatial transcriptomics is required to spatially locate each population and to predict their interactions with other cells, context that is crucial to fully define a population [30,31].

The BCSFB is emerging as a critical component of neurodegeneration and autoimmune disease development as well as in the restoration of homeostasis after stroke and trauma [32–34]. Due to its role in debris removal, a faulty BCSFB is implicated in the pathological accumulation that characterizes neurodegenerative diseases, including AD [32–34]. The BCSFB is more permeable than the BBB, leading intrathecal delivery to be a promising method of drug delivery to the CNS [35]. Yet, only a single study has reported on the use of FUS to promote intrathecal delivery via disruption of this barrier [36]. Despite being often within the area of treatment, this study provides the first investigation of the effect of FUS neuroimmunotherapy on the BCSFB [32–34].

Due to the role of brain macrophages in the neuroimmune response, they have been hypothesized to be a vital component of FUS neuroimmunotherapy mechanism. Most recently, our group reported the role of DAM and CAM in modulating the FUS-BBBO neuroimmunotherapeutic response in young and healthy animals [37]. The limited studies of brain macrophage response to FUS and FUS + MB neuromodulation indicate morphological activation in response to treatment [10,17–19].

This study provides the first comprehensive comparison of the three reported FUS neuroimmunotherapy paradigms by investigating the brain macrophage response in the presence and absence of Alzheimer's disease pathology. First, we use flow cytometry and

single-cell sequencing to demonstrate that FUS-BBBO maximizes macrophage modulation, including increases in phagocytosis and DAM across both genotypes. Next, we combine single-cell and spatial transcriptomics to analyze the complex genotype- and population-specific macrophage restructuring that occurs in response to FUS-BBBO with an overall trend toward increased association between brain macrophages and the BCSFB.

## 2. Materials and Methods

### 2.1. Mice

All procedures involving animals were approved by the Columbia University Institutional Animal Care and Use Committee. All mice were housed on a 12 h light/dark schedule with ad libitum access to food and water. Wild-type mice utilized female C57BL6/129 mice obtained from Jackson Laboratories. The 3xTg-AD line, which has been substantially characterized, was used as the Alzheimer's disease model [38]. All studies used female mice aged 11–12 mo.

For sequencing and validation experiments, n = 3/condition was chosen to maximize data while minimizing animal and monetary costs. For flow cytometry experiments, n = 6 was chosen to have sufficient statistical power due to minimal variation between samples and robust controls. Spatial sequencing experiments were performed on n = 1 section/condition due to monetary concerns. Validation of spatial transcriptomic was performed to ensure the robustness of the results with n = 3/condition.

#### 2.1.1. Focused Ultrasound Treatment

All mice (FUS-N FUS + MB, and FUS-BBBO) were anesthetized with oxygen and 1–2% isoflurane (SurgiVet, Smiths Medical PM, Inc., Dublin, OH, USA), placed on a stereotaxis (David Kopf Instruments, Tujunga, CA, USA) with head immobilized and depilated to reduce acoustic impedance mismatch. Identification of the suture was used for positioning of the transducer, as previously described [39]. A single-element, spherical-segment FUS transducer (center frequency: 1.5 MHz, focal depth: 60 mm, radius: 30 mm Imasonic, France) that was driven by a function generator (Agilent, Palo Alto, CA, USA) through a 50-dB power amplifier (E&I, Rochester, NY, USA) was used to treat the bilateral hippocampus. The center of the transducer held a pulse-echo ultrasound transducer (center frequency: 10 MHz, focal depth: 60 mm, radius 11.2 mm; Olympus NDT, Waltham, MA, USA) that was used for alignment and acquisition of cavitation data. The pulse-echo ultrasound transducer was driven by a pulse-receiver (Olympus, Waltham, MA, USA) connected to a digitizer (Gage Applied Technologies, Inc., Lachine, QC, Canada).

The transducer setup was attached to a three-dimensional positioning system (Velmex Inc., Lachine, QC, Canada). Each hippocampus was sonicated first for 10 s for cavitation baseline and then again for the experimental sonication. If utilized, microbubbles were injected intravenously between the baseline and experimental sonications. In all mice treated with MB (FUS-BBBO and FUS + MB), a bolus of 3 μL of lab-made microbubbles ($8 \times 10^8$/mL, mean diameter: 1.4 μm) was diluted with 100 μL sterile saline and then injected intravenously between the baseline and experimental sonications. The FUS + MB, FUS-BBBO, and FUS-N mice were treated with 250, 450 and 2 MPa FUS, respectively.

#### 2.1.2. Magnetic Resonance Imaging

Immediately following treatment, all FUS + MB, FUS-BBBO, and FUS-N animals underwent imaging with a 9.4T MRI system (Bruker Medical, Boston, MA). Mice were intraperitoneally injected with 0.2 mL of gadodiamide (OmniscanTM, GE Healthcare, Princeton, NJ, USA), which does not penetrate the intact BBB, exactly 30 min before scanning. Images were acquired using a contrast-enhanced T1-weighted 2D FLASH sequence (TR/TE 230/3.3 ms, flip angle: 70°, number of excitations: 6, field of view: 25.6 mm × 25.6 mm).

### *2.2. Tissue Processing*

2.2.1. Microglia Isolation

Exactly 2 h prior to sacrifice, animals were intraperitoneally injected with methoxy-XO4 (2 mg/mL in DMSO) at 5 mg/kg, consistent with the literature [40,41]. At the time of sacrifice, animals were transcardially perfused with 1× PBS until the liver demonstrated clearing (approximately 3–4 min). After perfusion, both hippocampi of each brain were extracted and combined prior to being homogenized. Cells were then filtered through a 70 μm filter, centrifuged, and prepared for myelin removal as specified by manufacturer's recommendations. Briefly, cells were resuspended in diluted MACS BSA stock solution (Miltenyi, 130-091-376) and incubated for 15 min with myelin removal beads (Miltenyi, 130-096-733) prior to being spun down and resuspended in diluted MACS BSA stock solution. The samples were then put through Miltenyi large separation columns (Miltenyi, 130-042-401) to deplete the myelin. After staining with relevant antibodies (specified in the table below), samples were spun down and resuspended in 1× PBS with DNAse (Worthington Biochemical, LS006343), RNAsin (Promega, PRN2615) and DAPI. Brain macrophages were then selected using the Influx Cell Sorter (BD).

Imaging Flow Cytometry

After selection via the Influx Cell Sorter, cells that underwent imaging flow cytometry were spun down and resuspended in 50 μL of 10% FBS prior to being run through the ImageStream (Luminex, Austin, TX, USA)). Image processing was performed with the (IDEAS 6.3, Austin, TX, USA).

Single-Cell Sequencing

Cells for sequencing were spun down post-selection and then resuspended at 1000 cells/μL in 10% FBS. Cells were immediately brought to the Columbia Single-Cell Sequencing Core, where viability was confirmed using a trypan blue exclusion assay. Cells were then loaded onto a single channel of a 3′ 10× Chromium chip for sequencing. The specifics of 10× single-cell RNA-sequencing have been well reported, but briefly, cells were partitioned into nanoliter-scale Gel Bead-In-EMulsions (GEMs), with each GEM producing cDNA with a common 10× barcode. This allows the reading of each transcript to be associated back with its original cell. The inclusion of genetic antibodies (hashing antibodies) allowed each cell to additionally be associated back with its original treatment. GEMs are incubated to produce barcoded, full-length cDNA that is amplified via PCR to achieve a sufficient size for library construction and preparation for library reading (Illumina, Inc., San Diego, CA, USA). Reading of the library on Ilumina produces a standard binary base call data output folder for downstream processing.

2.2.2. Immunohistochemistry

At the time of sacrifice, mice were transcardially perfused with 1× PBS until the liver demonstrated clearing (approximately 3–4 min). Brains were extracted and placed in 4% paraformaldehyde for 48 h of fixation prior to switching to 30% sucrose in 1× PBS until sectioning. Brains were cryostat sectioned at 35 μm. After sectioning, free-floating sections were washed with 1× TBS with 0.3% triton and blocked in a solution of 1× TBS with 0.3% triton and 5% donkey serum. Sections stained with lectin were then incubated with the lectin. Sections were then placed overnight at 4° in 1× TBS with 0.3% triton, 5% donkey serum, and the primary or conjugated antibody (concentration in table). Sections incubated with conjugated antibodies were washed with 1× TBS three times and then mounted with DAPI Mounting Medium (abcam, ab104139). Sections incubated with unconjugated primary antibodies were washed with 1× TBS and 0.3% triton three times and then incubated for 2 h with 1× TBS with 0.3% triton and 5% donkey serum and the secondary antibody concentrations indicated in the table. Tissues were then washed with 1× TBS three times and then mounted with DAPI Mounting Medium (abcam, ab104139).

*2.3. Data Analysis*

2.3.1. Cavitation Processing

Cavitation signals were received by a pulse-echo ultrasound transducer (center frequency: 10 MHz, focal depth: 60 mm, radius 11.2 mm; Olympus NDT, Waltham, MA, USA) that was used for alignment and acquisition of cavitation data. The pulse-echo ultrasound transducer was driven by a pulser-receiver (Olympus, Waltham, MA, USA) connected to a digitizer (Gage Applied Technologies, Inc., Lachine, QC, Canada). Prior to the injection of MB, 10 s of control pulses were taken to account for varying backgrounds between animals. A fast Fourier transform (FFT) was applied to each PCD pulse, and the resulting energy spectral density was bandpass filtered between the 3rd and 9th harmonics. The stable cavitation dose (SCD) was calculated as the log-fold change of the root-mean-square of the harmonic bins as compared to the root-mean-square the same values for the baseline signal [13–15] Table 1.

**Table 1.** Antibodies utilized for flow cytometry (FC) and immunohistochemistry (ICH) with their Targets, Concentrations, Companies and Reference numbers.

| Use | Target | Concentration | Company | Reference |
| --- | --- | --- | --- | --- |
| FC | CD11B-PE | 1:100 | Invitrogen | 12-0112-82 |
| FC | CD45-APC | 1:100 | Invitrogen | 17-0451-82 |
| IHC/FC | CD9-PE | 1:1000/1:100 | Biolegend | 124806 |
| IHC | CD169-PE | 1:200 | Biolegend | 142404 |
| IHC | NEUN-Cy3 | 1:500 | MilliporeSigma | MAB377C3 |
| IHC | IBA1-AF488 | 1:200 | MilliporeSigma | MABN92AF488 |
| IHC | Gt IBA1 | 1:500 | abcam | ab5076 |
| IHC | Rb IFITM3 | 1:1000 | Proteintech | 117141AP |
| IHC | Ms FOXJ1 | 1:1000 | Invitrogen | 14996582 |
| IHC | Dk anti-Goat FITC | 1:1000 | Invitrogen | A-11055 |
| IHC | Dk anti-Rb DY647 | 1:1000 | abcam | ab150079 |
| IHC | Dk anti-Ms TxRed | 1:1000 | Invitrogen | ab6818 |

2.3.2. Flow Cytometry

All flow cytometry post-processing was performed in FlowJo (FlowJo, LLC Ashland, Oregon, USA). Cells were first gated for size, doublets were removed, live cells were selected, and finally, brain macrophage count was quantified using *Cd11b*. Phagocytosis, *Cd9* and amyloid-beta levels were quantified as percentage of the alive brain macrophages.

2.3.3. Single-Cell Sequencing Analysis

Data Alignment

Barcoded reads were processed and aligned to the GRCm38 genome with Ensemble transcriptome annotation (GRCm38.p6) using CellRanger with default parameters.

Hashtag Demultiplexing

For single-cell sequencing runs, the barcoded hashtag reads were processed and aligned to the relevant antibody codes using CellRanger's default parameters. The unfiltered matrix was then loaded into R for subsequent processing. Cells with no antibody reads were removed, and then, the genetic and hashtag matrices were merged, only keeping UMIs present in both. Seurat functions were used to demultiplex the data and assign hashtags to each cell.

Quality Control and Integration

The data from the three $10\times$ runs were first filtered to only include cells between 2000 and 15,000 UMIs, less than 10% mitochondrial transcripts, and more than 1000 features as part of standard quality control. The data were then normalized for each run, and the top 3000 most variable features were identified. Finally, integration factors were found between all three runs, and the data were integrated using Seurat's default integration function.

Group Defining Gene Sets

The gene sets defining each treatment group, and cluster were found with Seurat's FindAllMarkers function on the raw gene expression data, which performs a differential expression test between the expression level of each gene in a single cluster versus the average expression in all other clusters.

### 2.3.4. Spatial Transcriptomics

Alignment

Raw FASTQ files were aligned to the GRCm38 genome with Ensemble transcriptome annotation (GRCm38.p6) and tissue locations using SpaceRanger with standard parameters [42].

Seurat Processing

Raw counts and alignment details for each section were loaded, integrated, and normalized in R using the Seurat Package with default parameters. Modified Seurat commands were used for the projection of parameters across each tissue slice [43].

Cell2Location

For prediction of cell location within our spatial transcriptomics sections, we created a custom reference by merging a downsampling of our single-cell sequencing macrophage clusters with a downsampling of publicly available mouse brain cell type data to mimic 200 cells per cluster or cell type [30,44]. Using Cell2Location with default parameters, we created a model to predict the location of all of these cell types within the tissue [30,44]. This model was then applied to each tissue section individually, and the resulting score for each cell type in each location was loaded into our R Seurat script for visualization. Pearson correlation and significance between cell types were found in R with the Hmisc package.

## 3. Results

### 3.1. FUS-BBBO Maximizes Brain Macrophage Response

In order to thoroughly compare the brain macrophage response to FUS-N, FUS + MB, and FUS-BBBO neuroimmunotherapy in the presence and absence of Alzheimer's disease pathology, we performed a combination of flow cytometry, imaging flow cytometry, and single-cell sequencing on cells from aged transgenic Alzheimer's animals (AD) and age-matched wild-type (WT) controls (Figure 1a). Four treatment groups were studied—naive, FUS + MB, FUS-BBBO, and FUS-N. Naive animals were anesthesia-, MB- and FUS-naive at the time of sacrifice. FUS + MB animals were intravenously injected with lipid-coated microbubbles (MB) and treated with FUS below the pressure threshold required for BBBO (250 kPa). FUS-BBBO animals were intravenously injected with MB and treated with FUS at the pressure for safe BBBO (450 kPa) [13,14]. The FUS-N group was treated with higher pressure FUS (2 MPa) without MB, consistent with the FUS-N literature [17–19]. All animals treated with FUS underwent a contrast-enhanced T1-weighted MRI, confirming BBBO in the FUS-BBBO animals and the absence of BBBO within FUS-N and FUS + MB animals Figure A1.

Prior to sacrifice, all animals were injected with methoxy-X04 to fluorescently label amyloid plaques for quantification of brain macrophage amyloid phagocytosis [40,41]. After transcardial perfusion, the bilateral hippocampus was dissected and prepared for flow cytometry selection of the alive brain macrophages. In addition to the macrophage marker *Cd11b*, cells were stained with *Cd9*, a marker of disease-associated microglia (DAM), and were incubated with pH-sensitive beads to measure phagocytosis levels. Naive macrophages from both WT and AD animals had similarly low levels of *Cd9+* and phagocytic cells. FUS-BBBO significantly increased the percentage of DAM (*Cd9+*) in both genotypes. FUS + MB also significantly increased the percentage of *Cd9+* brain macrophages in WT but not in AD animals. This increase is notably lower than the one triggered by FUS-BBBO (Figure 1b). FUS-BBBO increased the level of phagocytosis within both AD and WT cells, but only to a significant extent within our AD samples (Figure 1c). Finally, FUS-BBBO increased (non-significantly) the percentage of amyloid-positive brain

macrophage, representative of amyloid phagocytosis (Figure 1d). Overall, the FUS-N brain macrophages had no significant changes from the naive brain macrophages.

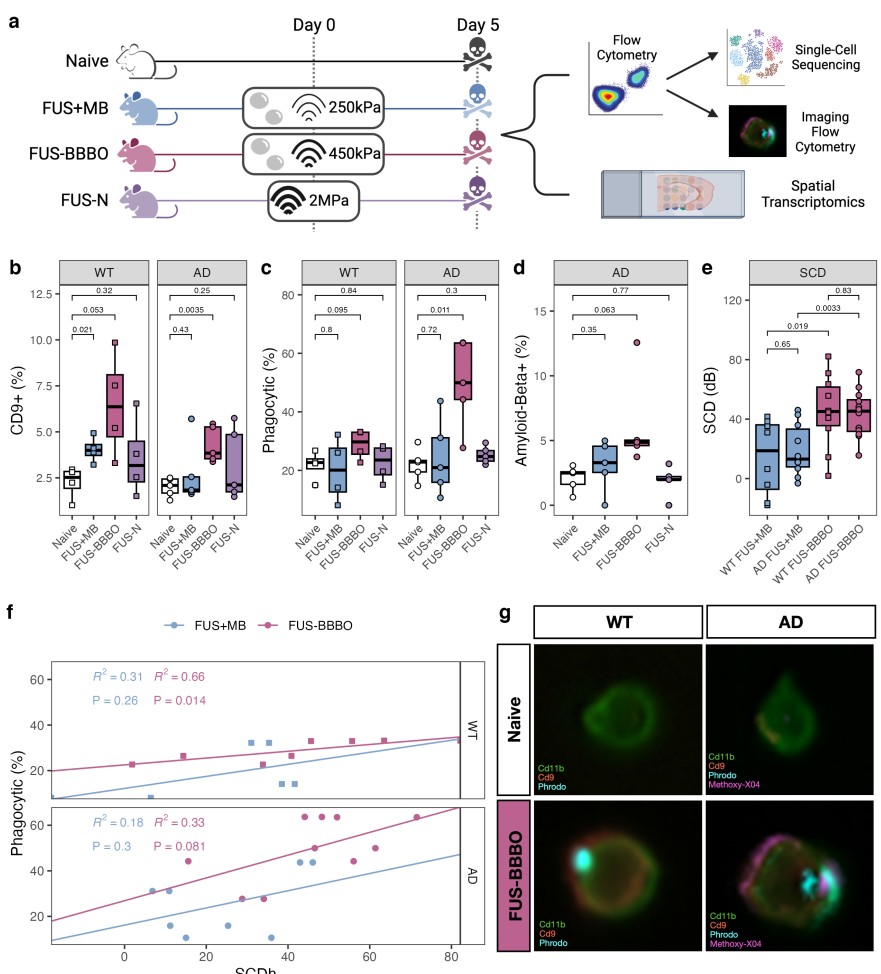

**Figure 1.** Focused ultrasound blood–brain barrier opening maximizes brain macrophage response. (**a**) Experimental schematic: mice were split into four groups—naive, FUS + MB, FUS-BBBO and FUS-N. Naive mice were anesthesia- and FUS-naive, FUS + MB mice were treated with MB injection and FUS at 250 kPa, too low to neuromodulate or induce BBB opening; FUS-BBBO mice were treated with MB injection and FUS at 450 kPa, safely and transiently opening the BBB; the FUS-N group was treated with 2 MPa FUS without MB to harness the thermal effects of ultrasound without BBB opening. Mice were treated on day 0 and sacrificed on day 5. Tissue was prepared for either flow cytometry selection and quantification of the brain macrophage or spatial transcriptomics. After selection, brain macrophages either underwent single-cell sequencing or imaging flow cytometry. (**b–d**) Flow cytometry comparison of the quantity of DAM (*Cd9+*), phagocytic and amyloid-beta+ brain macrophages between naive and the three paradigms of FUS neuroimmunomodulation in both aged transgenic Alzheimer's disease (AD) and age-matched wild-type (WT) animals. ANOVA followed by Bonferroni post hoc *t* tests were performed. (**e**) Stable cavitation dose (SCD) between FUS + MB and FUS-BBBO animals for both Alzheimer's disease (AD) and age-matched wild-type (WT) animals. ANOVA followed by Bonferroni post hoc *t*-tests were performed. (**f**) Correlation between SCD and phagocytosis for both FUS + MB and FUS-BBBO animals in both genotypes. Pearson regression analysis performed on each group. (**g**) Representative imaging flow cytometry from FUS-BBBO and naive brain macrophages from wild-type (WT) and Alzheimer's disease (AD) animals. Images illustrate the localization of *Cd11b*, *Cd9*, Phrodo and, in the case of the AD animal, amyloid-beta (flourescently tagged with Methoxy-X04).

As expected, at the lower pressure, FUS + MB has statistically lower SCD as compared to the animals treated with FUS-BBBO for both genotypes (Figure 1e). Furthermore, the

SCD was positively correlated with the percentage of phagocytic brain macrophages across both the FUS + MB and FUS-BBBO paradigms for both genotypes. This trend is more pronounced and is only statistically significant within the FUS-BBBO samples, even when animals experienced a similar SCD in the FUS-BBBO and FUS + MB groups. This indicates that SCD provides the optimal modulation of brain macrophage phagocytosis in the presence of BBBO (Figure 1e).

After the flow cytometry selection of the alive brain macrophages, cells underwent either single-cell sequencing or imaging flow cytometry. Imaging flow cytometry confirmed the location of proteins within the brain macrophages. *Cd11b* and *Cd9* were found on the cell surface, and the pH-sensitive beads and amyloid-beta were found to be internal (Figure 1f). Consistent with the process of macrophage phagocytosis and subsequent lysosomal digestion, the pH-sensitive beads and amyloid-beta colocalize [45].

Overall, the flow cytometry analysis of brain macrophages indicates that FUS-BBBO increases phagocytic and *Cd9*+ brain macrophages as compared to treatment with FUS + MB or FUS-N.

### 3.2. FUS-BBBO Alters Brain Macrophage Cluster Distribution

Three runs of single-cell sequencing were performed on the flow-cytometry-selected brain macrophages from each paradigm and genotype. Cells that passed preprocessing quality control were normalized and integrated across all runs [46]. After integration, cells were clustered with principal component analysis and were projected onto a Uniform Manifold Approximation and Projection (UMAP) dimensional reduction [43] (Figure 2a). Treatment with FUS-BBBO most drastically alters the subpopulation composition as compared to the three other treatment paradigms for both genotypes (Figure 2b).

To further characterize each brain macrophage cluster, we performed spatial transcriptomics on naive and FUS-BBBO tissue sections from both genotypes. Combining our single-cell and spatial transcriptomics with publicly available reference data (detailed in the Materials and Methods section), we were able to predict the spatial distribution and association of our brain macrophage clusters with other brain cells [30,44]. Coupling differential gene expression analysis (Figure 2c) with the predictions provided via the spatial transcriptomics analysis provided a robust characterization of each brain macrophage cluster (Figure 2d).

First, we identified Clusters 0, 1, 2, 3 and 5 as homeostatic due to their upregulation of homeostatic markers, including *Cx3cr1* and *P2ry12*. Clusters 0 and 3 were composed mostly of AD cells, Clusters 1 and 2 were composed mostly of WT cells, and Cluster 5 was a mix of both genotypes. Clusters 0, 1 and 5 were most strongly associated with neuronal subtypes (maximum correlation coefficients of 0.81, 0.82, and 0.75, respectively), consistent with the well-documented role of homeostatic microglia in synaptogenesis, synaptic plasticity, and neurogenesis [47,48]. Clusters 2 and 3 were most strongly associated with the brain–cerebrospinal-fluid-barrier (BCSFB) cells (maximum correlation coefficients of 0.79 and 0.65, respectively), consistent with the reported roles of brain macrophages in barrier surveillance and maintenance [49,50].

Next, we identified Clusters 4 and 6 as non-homeostatic due to their decreased presentation of homeostatic markers. Cluster 4 was further classified as our cluster of interferon-associated microglia (IAM), and Cluster 6 was further classified as our cluster of disease-associated microglia (DAM) by their upregulation of previously reported cluster defining genes [20,28,29,37]. The proportion of cells that are non-homeostatic increased in response to FUS-BBBO for both WT and AD samples. However, the increase in non-homeostatic brain macrophages is from an average of 4% in naive samples to an average of 31% for FUS-BBBO AD samples as compared to increasing from an average of 3% in naive samples to an average of 6% in FUS-BBBO WT samples. AD cells increase primarily in proportion of IAM as compared to WT cells, which increase equally in the representation of IAM and DAM. Both DAM and IAM have correlation coefficients less than 0.65 with other cell types within the brain.

Overall, we illustrated that FUS-BBBO alters brain macrophage cluster distribution in both WT and AD animals more than FUS + MB or FUS, and in response to FUS-BBBO, AD and WT cells increase their proportion of non-homeostatic DAM and IAM.

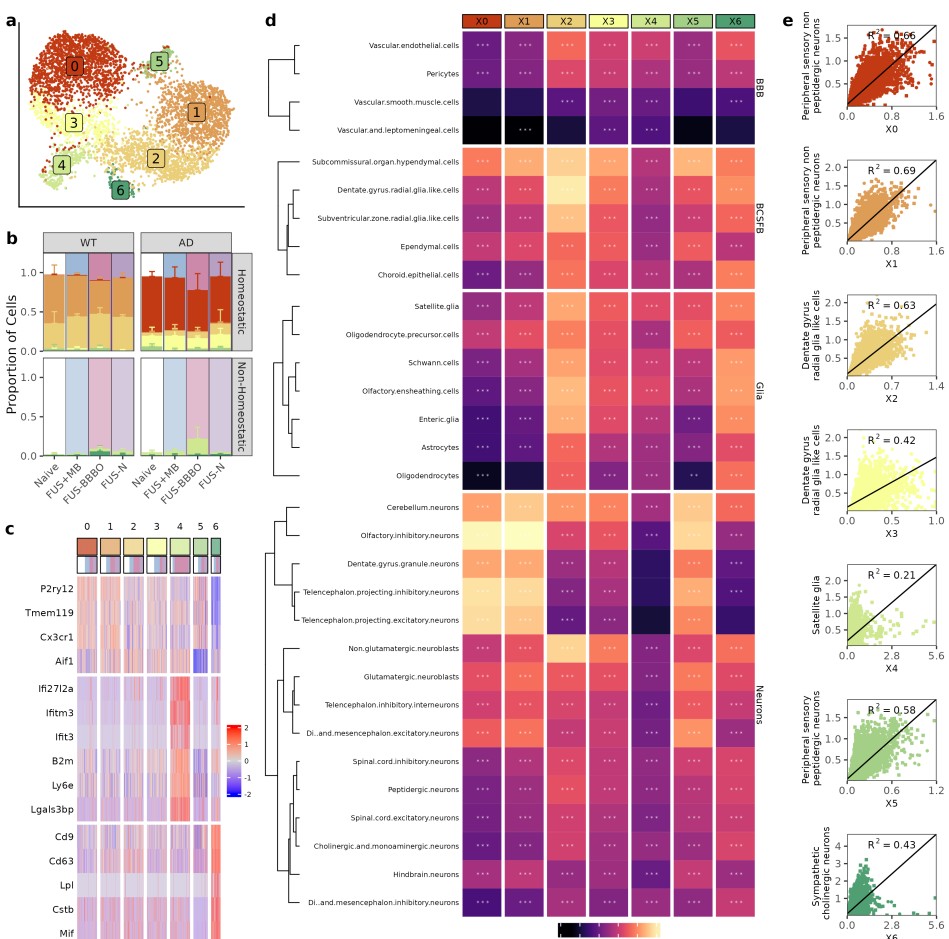

**Figure 2.** Brain macrophage clustering is defined by cell-type specific interactions. (**a**) UMAP of the brain macrophages after being selected, re-integrated and clustered. Each dot represents a cell and is colored based on its assigned cluster. (**b**) Distribution of clusters for each treatment group. Colors correspond to UMAP in (**a**). (**c**) Heatmap depicting the z-scores of the top differentially expressed genes across clusters. Cells were randomly downsampled to mimic equal distribution across clusters after marker selection. (**d**) Heatmap of correlation between macrophage cluster and all cell types. Cells are grouped based on type and clustered with k-means clustering. (**e**) Each macrophage cluster is shown with its strongest cell correlation (detailed in methods). Simple linear regression is shown on each plot (R2). ** $p < 0.005$, *** $p < 0.0005$. (**e**) Each macrophage cluster is shown with its strongest cell correlation (detailed in methods). Simple linear regression is shown on each plot (R2). The significance of correlation is displayed on each cell.

### 3.3. FUS-BBBO Alters DAM Cellular Associations

Based on prior reports by our group [37], and others [20,28], we identified our cluster of neuroprotective disease-associated microglia (DAM) by their upregulation of markers including *Cd9* and *Cd63* and downregulation of homeostatic microglia markers including *Cx3cr1* (Figure 3a,b). As predicted by their neuroprotective role, functional annotation of the DAM cluster-defining genes indicates roles in migration and phagocytosis (Figure 3d). Validating our flow cytometry analysis that found a genotype-independent increase in DAM (*Cd9*+) in response to FUS-BBBO, the proportion of sequenced brain macrophages that are DAM increases in response to FUS-BBBO for both AD and WT samples but more dramatically in WT samples (Figure 3d). Projecting the predicted DAM location across each spatial transcriptomics tissue section further confirms the FUS-BBBO-induced increase in DAM that is larger for WT animals (Figure 3e).

Furthermore, the association of DAM with other brain cells is also affected by FUS-BBBO. Naive WT and AD DAM have weak correlations with other brain cells. Naive WT

DAM are most strongly correlated with BCSFB cells (maximum correlation coefficient of 0.61), and naive AD DAM are most strongly correlated with glia cells (maximum correlation coefficient of 0.62). After FUS-BBBO, WT DAM correlations remain mostly unchanged, but AD DAM increases in correlation with BCSFB cells by 0.21, as well as increases in correlation with neurons by 0.35 (Figure 3f).

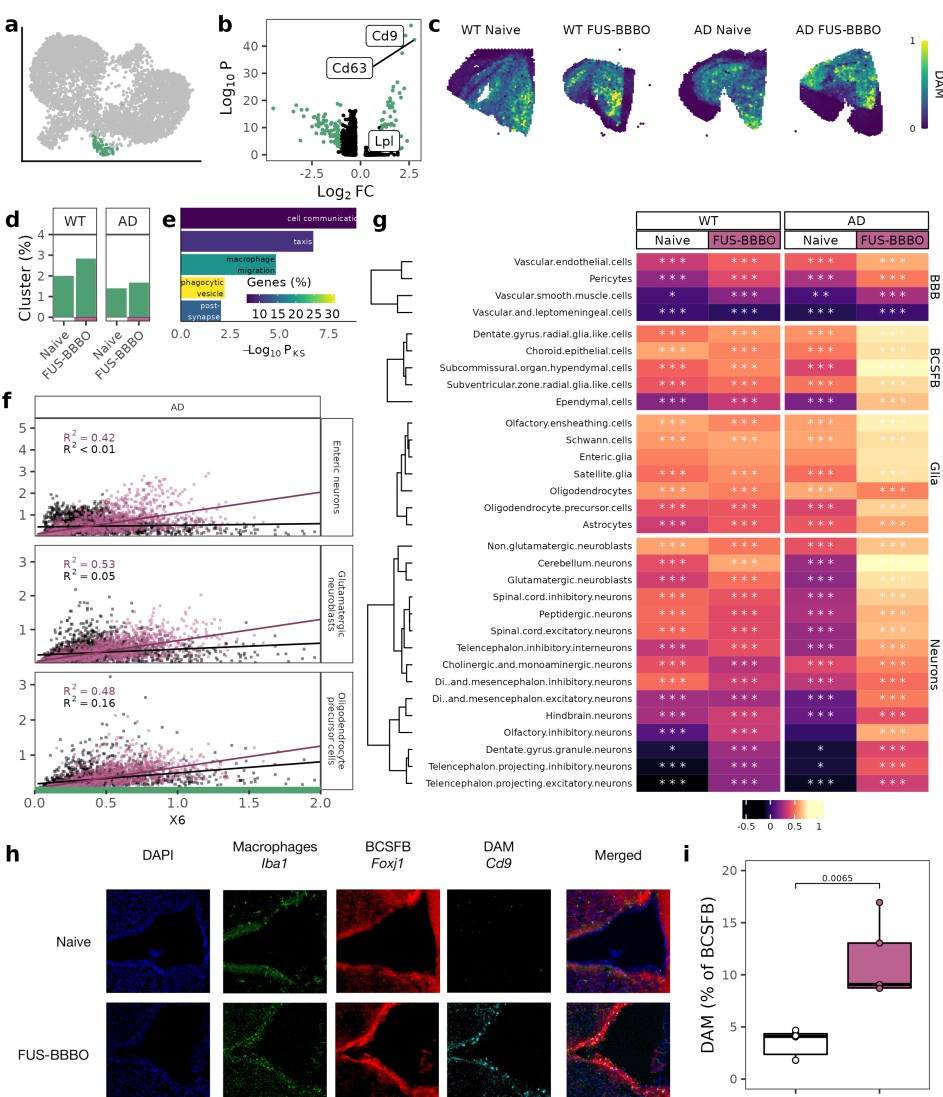

**Figure 3.** Focused ultrasound blood–brain barrier opening DAM interactions. (**a**) UMAP from 2a with DAM cluster highlighted. (**b**) Volcano plot comparing DAM cluster with all other brain macrophages with known DAM and homeostatic microglia markers labeled. Genes with a *p* value less than 0.05 and a log fold-change greater than 1 are in green. (**c**) DAM scores projected across each spatial transcriptomics tissue section. (**d**) Percentage of cells that are DAM for both genotypes and treatments. (**e**) Functional ontology of DAM defining genes. The Kolmogorov–Smirnov significance (Pks) is displayed on the x-axis, and the percentage of annotated genes that are significant is displayed in color. (**f**) Correlation between DAM and other select cells for both naive and FUS-BBBO AD samples. (**g**) Heatmap of correlation between DAM cluster score and other cell types split by genotype and treatment. Cells within each group are clustered with k-means clustering. Significance of correlation is displayed on each cell. (**h**) Representative immunohistochemistry for choroid plexus, with macrophages (*Iba1*) in green, ependymal cells (*Foxj1*) in red and DAM (*Cd9*) in cyan. (**i**) Quantification of immunohistochemistry images from (**h**). Two sections from three mice per group were stained and imaged. Quantification was performed on Matlab. Unpaired two-tailed *t* test was performed between groups. * $p < 0.05$, ** $p < 0.005$, *** $p < 0.0005$.

With immunohistochemistry, we are able to validate the increased interaction of DAM with the BCSFB in AD animals using DAM marker *Cd9* and BCSFB ependymal cell marker *Foxj1* (Figure 3h). Quantifying the immunohistochemistry, we see a statistically significant doubling of the association between DAM and the BCSFB in response to FUS-BBBO (Figure 3h).

### 3.4. FUS-BBBO Alters IAM Cellular Associations

Our IAM cluster, which has been documented to play a key role in cortical remapping, was identified by an upregulation of interferon genes (Figure 4a,b) [29]. Selecting, reintegrating, and clustering our IAM identified two sub-clusters: IAM 0 and IAM 1 (Figure 4c). These sub-clusters are identified by differential expression of a number of interferons and inflammatory markers, including *B2m* and *Ifitm3* (Figure 4e). The percentage of sequenced brain macrophages that are IAM increased for both WT and AD groups, but the increase is nearly ten-fold for the AD samples as compared to the lower two-fold increase in WT samples. Notably, the increase in IAM is an increase in both IAM 0 and IAM 1 cells for both genotypes (Figure 4d). Projecting the IAM clusters across the spatial transcriptomics tissue sections indicated that IAM 0 and IAM 1 increases are spatially distinct. IAM 0 increased evenly across the tissue, whereas IAM 1 increased heterogeneously near the hippocampus, the area treated with FUS-BBBO (Figure 4f,g). Using immunohistochemistry with macrophage marker *Iba1* and IAM 1 marker *Ifitm3*, we validated the heterogeneous increase in IAM 1 cells in the AD brain after FUS-BBBO (Figure 4i,j).

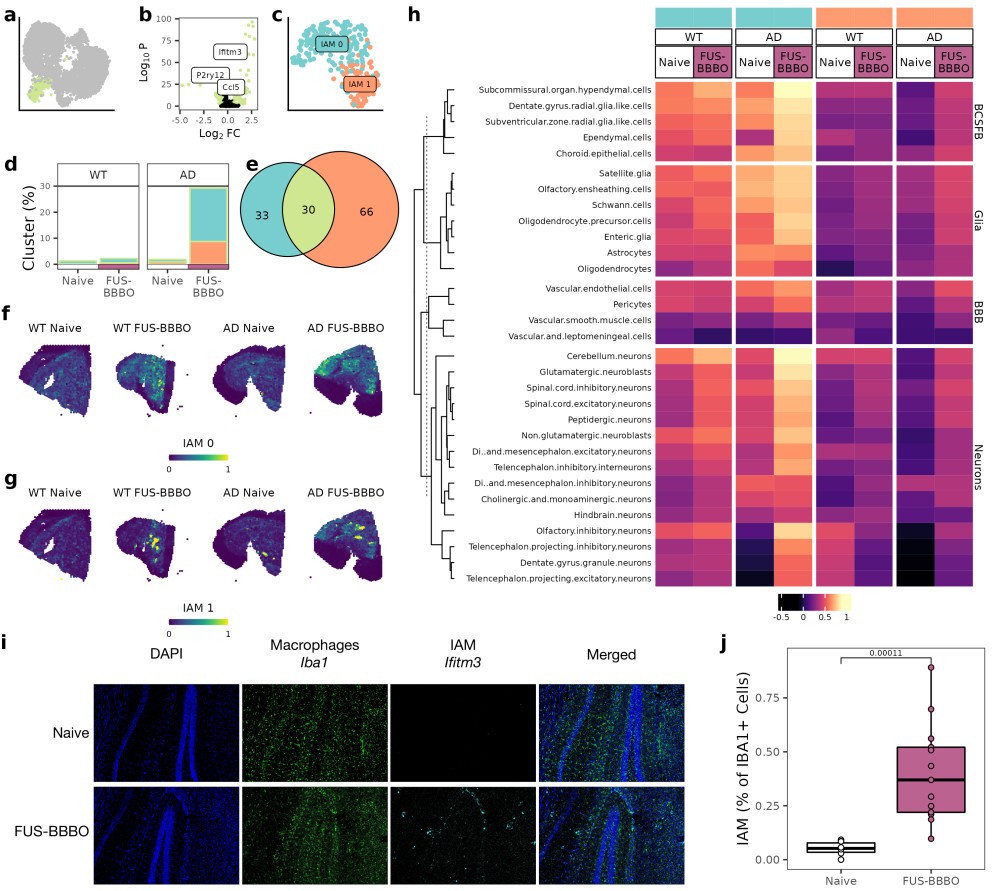

**Figure 4.** Focused ultrasound blood–brain barrier opening alters IAM interactions. (**a**) UMAP from 2a with IAM cluster highlighted. (**b**) Volcano plot comparing IAM cluster with all other cells with known IAM and homeostatic microglia markers labeled. Genes with a *p* value less than 0.05 and a log fold-change greater than 1 are in green. (**c**) UMAP of IAM subclusters after selection, reintegration

and clustering. (**d**) Proportion of cells that are IAM for both genotypes and treatments. IAM 0 and IAM 1 clusters shown with colors consistent to (**c**). (**e**) Comparison of cluster identifying genes for IAM 0, IAM 1 and all IAM. (**f**) IAM 0 projected across AD spatial transcriptomics sections. (**g**) IAM 1 projected across AD spatial transcriptomics sections. (**h**) Heatmap of correlation between IAM clusters and other cell types for both naive and FUS-BBBO AD sections. Cells within each group are clustered with k-means clustering. (**i**) Representative immunohistochemistry of AD hippocampus. Macrophages (*Iba1*) in green and IAM (*Ifitm3*) in cyan. (**j**) Quantification of immunohistochemistry images from (**i**). Two sections from three mice per group were stained and imaged. Quantification was performed on Matlab. Unpaired two-tailed *t* test was performed between groups.

For both WT and AD animals, FUS-BBBO increased IAM 0 interactions with BCSFB cells, but this increase is larger in the presence of disease. As a specific example, FUS-BBBO increased the IAM 0 and ependymal correlation coefficient from 0.59 to 0.71 in WT animals but from 0.35 to 0.79 in AD animals. FUS-BBBO affected IAM 1 interactions less; however, it slightly increased IAM 1 interactions with BCSFB and neurons across both genotypes (Figure 4h).

### 3.5. FUS-BBBO Alters CAM Cellular Associations

Our group previously reported that FUS-BBBO in young and healthy WT animals triggers an infiltration of CAM into the brain [37]. Using spatial transcriptomic analysis, we were able to localize the CAM within the brain and predict interactions with other cell types. Consistent with their role in barrier maintenance, we found that CAM in the naive WT brain are most strongly associated with BBB cells (maximum correlation coefficient of 0.88) (Figure 5a) [22,26]. This correlation is weaker but present in the naive AD brain, likely due to the AD-related increase in blood–brain barrier permeability [51].

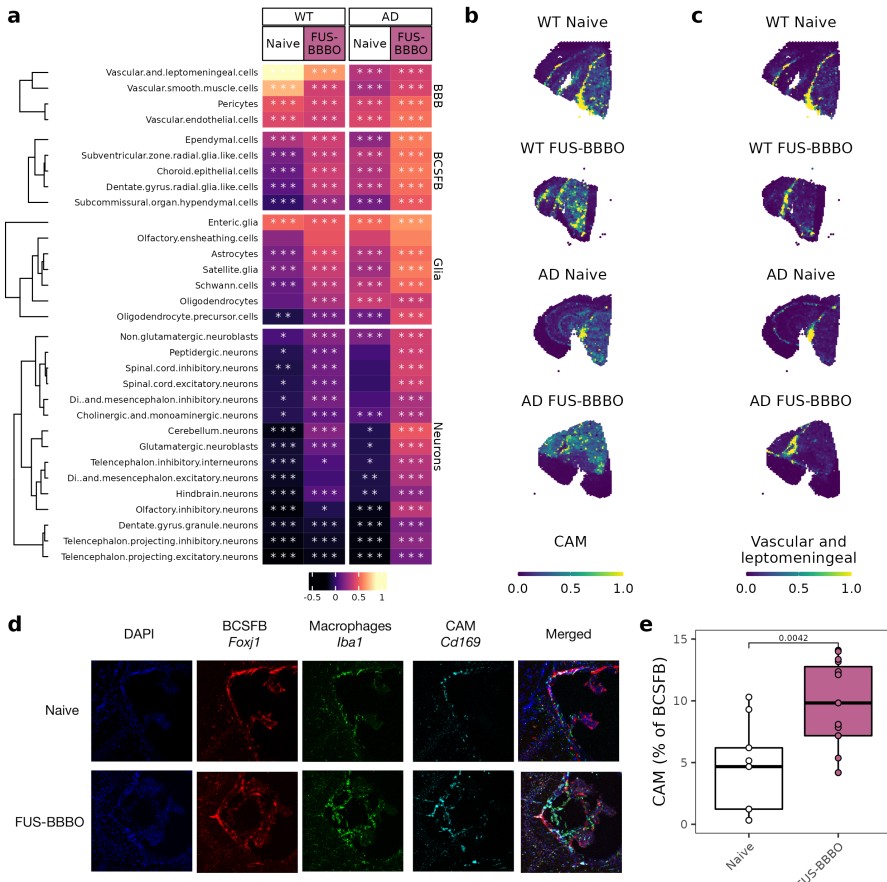

**Figure 5.** Focused ultrasound blood–brain barrier opening alters CAM location and interactions.

(**a**) Heatmap of correlation between CAM and other cell types for both genotypes and treatment conditions. Cells within each group are clustered with k-means clustering. Significance of correlation is displayed on each cell. * $p < 0.05$, ** $p < 0.005$, *** $p < 0.0005$. (**b**) CAM projected across WT and AD spatial transcriptomics sections. (**c**) Vascular cells projected across WT and AD spatial transcriptomics sections. (**d**) Representative immunohistochemistry of AD choroid plexus. Macrophages (*Iba1*) in green, CAM (*Cd169*) in cyan and ependymal cells (*Foxj1*) in red. (**e**) Quantification of immunohistochemistry images from (**d**). Two sections from three mice per group were stained and imaged. Quantification was performed on Matlab. Unpaired two-tailed *t* test was performed between groups.

After FUS-BBBO, WT and AD CAM respond in a genotype-specific manner. WT CAM decrease association with BBB cells and moderately increase association with all other cell types. AD CAM, however, slightly increase association across cell types but most strongly with BCSFB cells. As a specific example, AD CAM increase in correlation with ependymal cells from 0.08 to 0.38 in response to FUS-BBBO. With immunohistochemistry, we are able to validate the increased interaction of AD CAM with the BCSFB using CAM marker *Cd169* and BCSFB ependymal cell marker *Foxj1* (Figure 5d,e).

## 4. Discussion

Due to being entirely noninvasive and safe, focused ultrasound (FUS) has emerged as a promising method of neuroimmunotherapy for the treatment of a number of neurological disorders [3,4,6,9]. However, the neuroimmunotherapeutic potential remains poorly optimized, and its mechanism remains to be comprehensively characterized. This study addresses these limitations by first comparing the brain macrophage modulation to three distinct focused ultrasound paradigms, identifying focused ultrasound blood–brain barrier opening (FUS-BBBO) as the paradigm that maximizes modulation and, secondly, combining single and spatial transcriptomics to detail the complex brain macrophage redistribution that occurs in response to treatment.

Expanding on a recent report by our group that characterized the FUS-BBBO-induced brain macrophage modulation in young and healthy animals, we compared the efficacy of FUS-BBBO, FUS-N, and FUS + MB in modulating the brain macrophage response in the presence and absence of disease pathology [2,10,17,19,37,52]. We demonstrated that FUS-BBBO maximizes brain macrophage phagocytosis and subpopulation composition shifts as compared to the two other paradigms. We did not find any significant effects of FUS-N in modulating the brain macrophages, which is unlike what has been previously shown [17,19]. We hypothesize that this discrepancy is due to the number of treatment sessions—our study investigated the effect of a single treatment session, whereas the FUS-N studies reported macrophage modulation after multiple treatment sessions [17,19].

In the second part of the study, we combined single-cell and spatial transcriptomics with immunohistochemical validation to localize brain macrophage subpopulations within the naive and FUS-BBBO-treated brain, illustrating that FUS-BBBO restructures macrophage associations. More specifically, we illustrated that disease-associated microglia (DAM), interferon-associated microglia (IAM), and central-nervous-system-associated macrophages (CAM) increase association with blood-cerebrospinal-fluid-barrier (BCSFB) cells and neurons in response to FUS-BBBO. These shifts in interaction are larger in the presence of AD pathology, indicating a greater effect of treatment in the presence of disease. The overall increase in brain macrophage activation is consistent with reports of morphological activation and phagocytosis [3,5,9,37,53]. Additionally, the increased interaction with neurons is consistent with reports of FUS-BBBO-induced neurogenesis and the role of brain macrophages in neurogenesis [8,37,47,48,54–56].

To the best of our knowledge, despite often being within the area of the FUS treatment, no studies have reported the effect of FUS-BBBO on the BCSFB. The BCSFB has an emerging significance in brain immune response and neurodegeneration development. In response to perturbation, the BCSFB can recruit peripheral cells and increase the transport of debris out of the brain [32,33]. In neurodegeneration, however, the BCSFB fails to

eliminate disease pathology from the brain [33,34]. Due to the documented localization of DAM and IAM around both areas of perturbation and disease pathology, there are two interpretations of the brain macrophage localization to the BCSFB after FUS-BBBO. The first is that FUS-BBBO may perturb the BCSFB and the DAM and IAM function to restore homeostasis [20,28,29]. The second is that brain debris is cleared through transport across the BCSFB, a process that becomes dysfunctional in neurodegeneration [32,33]. It is possible that the documented reduction of pathology in response to FUS-BBBO is due to restoration of this process, and DAM and IAM support this process by surrounding the pathology as it is transported [32,33]. Future work will determine which of these hypotheses are valid regarding the localization of the populations to the BCSFB.

FUS-BBBO is currently being translated to the clinic as both a method of drug delivery and a neuroimmunotherapeutic [18,57,58]. Optimal adoption of the technique requires paradigm optimization and mechanism identification. This study addresses these limitations by directly comparing three methods of FUS neuroimmunotherapy and by utilizing bioinformatics to localize brain macrophage modulation in response to treatment. The major limitation of this study was that a single time-point and treatment session was evaluated for all studies due to the temporal and potentially compounding effect of FUS neuroimmunotherapy treatment. Future work will include an evaluation of the temporal and compounding effects of treatment.

Overall, FUS-BBBO was proven to be the most efficacious method of FUS brain macrophage modulation and remains a promising method of neuroimmunomodulation.

**Author Contributions:** Conceptualization, E.E.K.; Validation, A.R.K.-S.; Formal analysis, A.R.K.-S.; Investigation, A.R.K.-S.; Resources, E.E.K.; Data curation, A.R.K.-S. and R.L.N. and H.P.; Writing—original draft, A.R.K.-S.; Writing—review and editing, A.R.K.-S. and E.E.K.; Visualization, A.R.K.-S.; Supervision, V.M. and E.E.K.; Project administration, E.E.K.; Funding acquisition, E.E.K. All authors have read and agreed to the published version of the manuscript.

**Funding:** This research was funded by the National Institutes of Health R01AG038961 and National Science Foundation DGE-2036197.

**Data Availability Statement:** All data and codes used for analysis are publicly available. Raw and processed spatial and single-cell sequencing data are available on GEO (Number TBD), and codes are available on https://github.com/ark2173/MG2.0.

**Conflicts of Interest:** The authors declare no conflict of interest.

**Abbreviations**

The following abbreviations are used in this manuscript:

| | |
|---|---|
| BBB | Blood–brain barrier |
| FUS | Focused ultrasound |
| MB | Microbubbles |
| FUS-BBBO | Focused ultrasound blood–brain barrier opening |
| FUS-N | Focused ultrasound neuromodulation |
| FUS + MB | Focused ultrasound and microbubbles |
| AD | Alzheimer's disease |
| WT | Wild-type |
| BCSFB | Brain–cerebrospinal fluid barrier |
| SCD | Stable cavitation dose |
| DAM | Disease-associated microglia |
| IAM | Interferon-associated microglia |
| CAM | Central-nervous-system-associated macrophages |
| UMAP | Uniform Manifold Approximation and Projection |

**Appendix A**

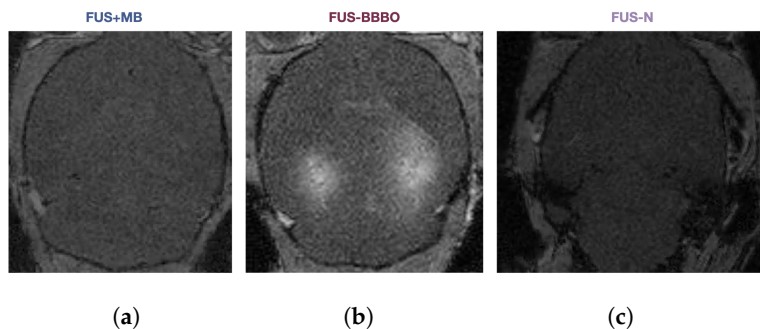

(**a**)          (**b**)          (**c**)

**Figure A1.** Representative gadodiamide–MRI image from (**a**) FUS + MB, (**b**) FUS-BBBO, and (**c**) FUS-N treatment groups.

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
