# Peer review of "Focused Ultrasound-Mediated Blood–Brain Barrier Opening Best Promotes Neuroimmunomodulation through Brain Macrophage Redistribution"

_2571-6980, doi:10.3390/neuroglia4020010_

Round 1

Reviewer 1 Report

The manuscript by Kline-Schoder et al. discusses the use of focused ultrasound (FUS) as a neuroimmuno-therapeutic for neurological diseases. FUS can be used in three paradigms: FUS-BBBO, FUS-N, and FUS+MB. The study focuses on the role of brain macrophages in response to the three paradigms and compares the modulation of macrophages in the presence and absence of Alzheimer's disease pathology. The study uses flow cytometry and single-cell sequencing to identify FUS-BBBO as the most effective paradigm for modulating brain macrophages. The study also analyzes the complex genotype- and population-specific macrophage restructuring that occurs in response to FUS-BBBO using single-cell and spatial transcriptomics. Overall, the study provides insight into the mechanism of FUS neuroimmunomodulation and identifies FUS-BBBO as the optimal paradigm for modulating brain macrophages.

In summary, I think the quality of this manuscript is suitable to publish on Neuroglia in the current form. I think the author can improve the manuscript by adding an illustration/figure combing FUS device and mice to let reader know how the technique function.

Reviewer 2 Report

---In this manuscript, the authors assessed and compared the modulation of brain macrophages following FUS-BBBO, FUS-N, and FUS+MB. In addition, they investigated the effect of FUS-BBBO on brain macrophage redistribution utilizing single- and spatial transcriptomics. The authors report that compared to FUS+MB and FUS, the effect of FUS-BBBO on brain macrophages distribution is more pronounced in both WT and AD mice, an effect that is associated with increased levels of DAM and IAM, which were associated with BCSFB. The manuscript provides new findings; however, additional details and clarifications are required. The authors presented the data and figures; however, there needs to be a discussion on what it means. I have the following comments:

 --- A discussion on what it means for DAM and IAM increased expression following FUS-BBBO and the significance of their redistribution to BCSFB. How this would impact the pathology, especially following acute exposure of a single dose compared to multiple exposures.

--- The introduction could benefit from additional information on the different studied types of macrophages and their alteration by AD pathology.

--- The authors conducted a gadodiamide-MRI experiment; however, no data was shown.

--- Details on the age, sex (female mice??), and the number of mice used per experiment should be reported with justification for the selected age and sex. In addition, to help repetition by others and for reproducibility, details are necessary on the time frame of some analyses before animals’ sacrifices following the injection of gadodiamide and methoxy-X04.

--- The authors should revise the figures and their legends as they often do not match their labels. For example, Figure 5 shows a-d, but the legend describes a, b, c, I, and j; there are two “e” in Figure 1, and so on…

---   The data described on page 13 (lines 435-440) do not reflect the cited figure. Clarification is required.

Round 2

Reviewer 2 Report

Adding a representative gadodiamide-MRI image from each group as a supplementary figure is helpful; however, no supplementary file is found. Please add the supplementary figure with its legend.

In addition, Figure 1 continues to have 2 figures labeled with "e" while the figure legend includes "g". Please review all figures and legends carefully.
